# Dietary Inclusion of Defatted Silkworm (*Bombyx mori* L.) Pupa Meal for Broiler Chickens at Different Ages: Growth Performance, Carcass and Meat Quality Traits

**DOI:** 10.3390/ani13010119

**Published:** 2022-12-28

**Authors:** Eszter Zsedely, Marco Cullere, Georgina Takacs, Zsolt Herman, Klaudia Szalai, Yazavinder Singh, Antonella Dalle Zotte

**Affiliations:** 1Department of Animal Science, Szechenyi István University, Egyetem tér 1, H-9026 Győr, Hungary; 2Department of Animal Medicine, Production and Health, University of Padova, Agripolis, Viale dell’Università 16, Legnaro, 35020 Padova, Italy

**Keywords:** hybrid chicken, insect, feeding, meat physical traits, meat proximate composition

## Abstract

**Simple Summary:**

In the last decade, the use of insects as feed ingredient has been a popular research topic worldwide. In Europe, a strong input has occurred since the new regulation allowing the use of animal protein to feed non-ruminants, as well as the recent evolution of the market concerning feed ingredients. Insects represent a sustainable feed ingredient alternative to commercial feedstuffs, with enormous potential thanks to a rich nutritional profile, and the possibility to reduce feed-food competition. Among different species, silkworm (*Bombyx mori*) is of interest thanks to the protein amount, quality, and healthy lipids of spent pupae. Despite the potential, further research on this insect is required to determine the suitability for different livestock species, optimum inclusion levels, administration periods, and possible undesired side-effects. The present study contributes to providing a better understanding of the potential of silkworm as feedstuff for broiler chickens.

**Abstract:**

The present study was conducted to assess the effects of a 4% defatted silkworm (SWM-DEF) meal dietary incorporation into chickens’ diet at different growth stages on growth performances, carcass, and meat quality traits. A total of 90 Ross 308 one-day-old male broiler chickens were randomly allocated into 3 dietary groups of 5 replicated pens/diet (6 chickens/pen). One group was fed a standard soybean-based diet (C); group SWM1 consumed a starter diet (1–10 days of age) including 4% SWM-DEF and then the C diet up to slaughter (11–42 days of age); group SWM2 was fed with the C diet in the starter phase and the 4% SWM-DEF diet up to slaughter. Individual live weight and pen feed intake were determined at the end of each feeding phase: starter, grower, and finisher. Weight gain and FCR were then calculated. At 42 days of age, chickens were slaughtered and carcass traits determined. Leg and breast physico-chemical meat quality was also evaluated. Results confirmed that SWM-DEF could be a possible alternative feed source for chickens since growth performance, carcass, and meat physical traits were overall comparable in the three treatment groups. The feeding stage seemed to play a relevant impact on the sole meat protein content (SWM1 < SMW2 and C; *p* < 0.001). Concluding, the results available to date indicate that 4% SWM-DEF can be provided to chickens in different growth phases, and the choice of the inclusion period is more an industrial cost-benefit evaluation. From the scientific point of view, however, an administration in the grower-finisher phase rather than in the starter one provided the best meat nutritional quality. Further investigations should focus on the impact on meat fatty acids profile and sensory traits, which are of utmost importance for consumers.

## 1. Introduction

The use of insects as an innovative and sustainable feed source for food-producing animals has been a popular research topic worldwide in the last decade [1]. In the European context, the new commission regulation 2021/1372 of 18 August 2021 amended Annex IV to Regulation (EC) No 999/2001 of the European Parliament and of the Council, allowing the use of insect processed animal proteins (PAP) in pig and poultry feed [2]. This is expected to give further impetus to the European production sector and, consequently, related research. Insects and derived products (i.e., protein meal and fat/oil) showed to be promising non-conventional animal feed for poultry, pig, and aquaculture species [3,4,5,6,7,8]; however, the insect sector still has many challenges to solve linked to product development, including the choice of the most suitable species as feed for selected animal species, optimal inclusion levels, as well as product safety, healthiness, and economic sustainability [6,7,9,10,11,12], which motivate further research.

Among insects, one species of possible interest is the silkworm (*Bombyx mori* L.), whose pupae are the byproduct of silk production and, at present, represent an environmental issue as they are mainly discarded as waste, despite their nutritional value [4,13,14]. In fact, the pupa can be processed to obtain oil and protein meal (SWM): SWM could be a potential protein source in chicken farming thanks to its high content of protein (60–75%, even higher than soybean or fishmeal) of high biological value [13]. Its oil is rich in omega-3 fatty acids, mainly linolenic acid (C18:3 *n*-3), known to be beneficial for human health [12].

Up to now, available research on the SWM as poultry feed is still scarce compared to that conducted on the most common insect species such as *Hermetia illucens, Tenebrio molitor*, or *Musca domestica*: key aspects for an emerging feed ingredient such as those linked to species-specific optimum inclusion levels and inclusion period (starter, grower, finisher stage of the whole farming cycle) remain to be investigated. Furthermore, published results concerning SWM utilization in poultry diets still leave some gaps: in some studies, no negative effects on chicken performance were noticed [14,15,16,17], while in other studies some possible critical points related to SWM utilization have been highlighted. Specifically, Miah et al. [14] highlighted that the inclusion of either 7% or 14% of defatted SWM in the diet of chickens was not optimal as both inclusion levels increased feed intake and reduced the protein content of meat. Dalle Zotte et al. [18] hypothesized that the 1-Deoxynojirimycin, an alpha-glucosidase inhibitor which is commonly found in mulberry leaves (main feed source of the silkworm), was responsible for the observed impairment of the apparent nutrients digestibility of meat-producing quails fed either with 12% defatted or full-fat SWM. Another critical point, common with most insect species including silkworm, is chitin content, a structural polysaccharide made from chains of modified glucose that negatively affects nutrient utilization in poultry [19].

Based on these premises, it is necessary to establish if the SWM can effectively be considered a possible feed ingredient for poultry diets and, to shed light on this aspect, it is fundamental to identify an inclusion level that provides satisfactory results. Moreover, it is important to find out if the inclusion period (i.e., growth stage of the animal) plays a role on the productive outcomes and product quality.

Therefore, the first objective of this study was to determine whether a 4% dietary inclusion of a defatted SWM (SWM-DEF) can be considered a realistic inclusion level for broiler chickens, as a partial replacement of soybean meal. A second objective of the research was to assess if the growth stage of the chicken is a factor that needs to be considered for choosing the SWM-DEF dietary inclusion period. For the evaluation of the above-mentioned aspects, chickens’ growth performance, carcass traits, and meat physico-chemical traits were considered.

## 2. Materials and Methods

### 2.1. Experimental Design and Bird Management

The experiment was carried out at the Animal Research Plant of Szechenyi Istvan University (Hungary). All birds were handled according to the principles stated in the European directive 2010/63/EU on the protection of animals used for scientific purposes and the 2021/1372/EU, and according to the Hungarian legal requirements 32/1999./III. 31./and 178/2009./XII. 29./. For the present experiment, approval by the local ethical committee was not deemed necessary, since chickens were fed diets based on non-toxic ingredients and no invasive procedures were performed on living animals.

A total of 90 one-day-old Ross 308 male broiler chickens were purchased from Gallus Poultry Farming and Hatching Ltd. (Hungary). Chicks were vaccinated against Newcastle disease, infectious bronchitis, infectious bursal disease, and fowl poxvirus. Once in the hatchery, chicks were individually identified with a wing band and randomly divided into three dietary groups of 5 replicates/each (6 chicken/replicate). Each replicate consisted of a pen of 4 m^2^ (1.5 bird/m^2^), equipped with 5 cm depth wood shavings as bedding material. Water and feed were provided *ad libitum* by bell automatic drinker and trough feeder, respectively. Feed formulation, temperature, and lighting schedule were set in accordance with the AVIAGEN Ross 308 management guide [20]; temperature and daylight period were regularly checked during the whole experiment.

The first group of chickens received a control diet (commercial corn-soybean based poultry feed; C) throughout the growing period of 42 days, the second group received a diet including 4% defatted silkworm meal (SWM-DEF) during the starter phase (1–10 days) (SWM1) and the C diet up to slaughter, and the third group was fed with the C diet in the starter phase and the 4% SWM-DEF diet (grower and finisher) up to slaughter (SWM2). Experimental diets were formulated to be isonitrogenous and isoenergy, based on digestible amino acids and metabolizable energy. As for SWM-DEF amino acid digestibility was unknown, calculated values based on the database system of Schothorst Feed Research (The Netherlands) were used. Diets were formulated by Agrofeed Ltd., (Győr, Hungary) and manufactured at the Animal Research Plant of the University. The ingredients of the experimental diets are reported in Table 1.

During the trial, individual live weight of chickens was measured four times: at the beginning of the trial (1ST day of age) and after each dietary period (starter, 10th day of age; grower, 25th day of age; finisher, at a 42nd day of age). For each pen, feed intake was registered in each dietary period. Individual weight gains and pen feed conversion ratio (FCR) were then calculated. Chicken health status and mortality were daily monitored.

### 2.2. Chemical Composition of Defatted Silkworm Meal, Soybean Meal and Experimental Diets

Defatted silkworm meal, soybean meal and experimental diets were analyzed for moisture, crude protein, crude fiber, ether extract, and ash contents following the AOAC methods [21]. The same analytical procedures were followed to determine the proximate composition of SWM-DEF and soybean meal. Chitin content of the experimental diets was analyzed in triplicate following the method provided by Zhang and Zhu [22] with the modifications described by Woods et al. [23]. The lysine and methionine contents of the SWM-DEF, soybean meal, and experimental diets were analyzed in accordance with ISO 13903:2005 [24]. Gross energy (GE) of SWM and soybean meal was analyzed with an adiabatic bomb calorimeter (ISO, 1998) [25]. The chemical composition of SWM-DEF, soybean meal, and experimental diets is presented in Table 2.

### 2.3. Slaughter, Carcass Dissection, and Samples Preparation

At 42 days of age and after overnight fasting, 15 chickens/treatment (3 chickens/replicate) were weighed and transported to a commercial abattoir located 20 km away from the Animal Research Plant and slaughtered following the standard commercial procedure (Council Regulation, EC, No. 1099 of 2009). Birds were electrically stunned (120 V, 200 Hz), bled, soft-scalded (56 °C for 2 min), plucked, and eviscerated. Carcasses were chilled at 4 °C and stored at the slaughterhouse until the following day. Thereafter, carcasses were transported (+4 °C) at the Szechenyi Istvan University Animal Department, Hungary and here they were divided into two halves: one was the sample dedicated to meat quality evaluations (carcass weight and yields, ultimate pH (pHu), drip, thawing and cooking losses, breast and leg meat proximate composition) at the Szechenyi Istvan University Animal Department, and the second half of each carcass was frozen (−20 °C) and transported to the Department of Animal Medicine, Production and Health (MAPS) of the University of Padova, Italy (leg physical measurements and further meat quality analyses, not included in the present study).

The same meat samples used to compute thawing loss were used for cooking loss determination.

### 2.4. Meat Quality Evaluations

After carcass dissection (*n* = 45: 15 half-carcasses/treatment), chicken breast and thigh were excised. The pHu was measured with a portable one-hand pH/°C meter equipped with a glass electrode (Testo SE & Co KGaA, Celsius straße 2, 79,822 Titisee-Neustadt, Germany) by completely inserting the electrode into the superior portion of the breast (*Pectoralis major* muscle) and thigh (*Biceps femoris* muscle) [26].

For drip loss determination, fresh meat samples of 80 g and 40 g were excised from breast (caudal part of *Pectoralis major* muscle, triangular shape) and thigh (*Biceps femoris*, rectangular form), respectively, and placed into an inflated bag avoiding the contact meat-bag. Samples were then suspended and stored at 4 ± 0.7 °C (Dixell Cool mate refrigerate room) for 24 h storage. Afterwards, samples were gently blotted dry and again weighed to compute drip loss [27].

For thawing loss determination, meat samples of 50 g and 13 g were excised from breast (middle part of *Pectoralis major* muscle, rectangular form) and thigh (*Biceps femoris*, rectangular form), respectively, and placed into a food-grade plastic bag and stored in a freezer (Zanussi Lehel, Lehel Refrigeration Factory Ltd., Type: CF 400) at −20 °C for 3 months. Afterwards, samples were removed from the freezer and put in a refrigerator (at 4 ± 0.7 °C) for 24 h to allow thawing. Samples were then removed from the bag, blotted dry, and weighed again.

For cooking loss determination, samples were placed into food-grade plastic bags, vacuum-sealed, immersed into water-bath (Premium Sous Vide Digital Water Bath SV25, Instanta Southport, Merseyside England) set at 80 °C, and cooked for 30 min. Samples were then removed from the water bath and cooled before measuring the weight. Meat samples were then freed from bags, blotted dry, and weighed (W6) [27].

After sample excision for drip and thawing loss determinations, the remaining meat of fresh breast and leg meat cuts was dedicated to the proximate composition analysis, which was carried out following the AOAC methods [21].

At the meat quality lab of the MAPS Department (University of Padova, Italy), a total of *n* = 30 (10 half-carcasses/treatment) were selected and thawed overnight at +4 °C. Deboned leg and its parts (meat, bones, skin, dissectible fat) were weighed after drying with paper towel to calculate yields and meat to bone ratio. The femur bone was removed, cleaned, and dedicated to Warner–Bratzler Fracture Toughness (WBFT) measurement: it was performed at the average bone length point using a dynamometer Texture TA-HD (SMS—Stable Micro System) with a 6 cm wide cell and a load rate of 0.5 mm/s [28].

### 2.5. Statistical Analysis

Experimental data were subjected to the one-way ANOVA with experimental diets (Control, SWM1, and SWM2) as a fixed effect and following the general linear model (GLM) procedure of SAS 9.1.3 statistical analysis software for windows (SAS, 2008). Cage was considered as random effect. For growth performance, the single chicken was the experimental unit; for FI and FCR, the experimental unit was the pen; and for carcass and meat physicochemical traits, the experimental unit was the single carcass. Least square means were obtained and post-hoc pairwise comparison was performed using the Bonferroni correction. The significance was considered at 5% confidence level.

## 3. Results

### 3.1. Live Performance and Carcass Traits

Results on the effect of a 4% SWM-DEF dietary inclusion in different periods of broilers’ productive cycle on live performance are depicted in Table 3 and Table 4. Independently to the SWM-DEF inclusion and the inclusion period, chickens of all the experimental groups always showed comparable live weight and weigh gains (*p* > 0.05). During the trial, chickens displayed an optimal health status and only one SWM2 chicken died throughout the experiment.

Coherently to what it was observed for live weight and weight gains, the dietary SWM-DEF inclusion did not affect feed intake and FCR which were similar in the three groups of chickens (*p* > 0.05). The average FCR was 1.97 in the starter, 1.38 in the grower, and 1.89 in the finisher phases, for an overall average value of 1.75.

Results presented in Table 5 overall showed that the dietary inclusion SWM-DEF (either SWM1 or SWM2) did not modify the considered carcass traits. Carcass yield was in the range 68.3–69.2%, the average breast yield was 30.4%, while the average leg yield was 29.5%. Similarly, leg dissection yields were not affected by the dietary inclusion of a 4% SWM-DEF either in the starter or in the grower-finisher periods: the average leg weight was 276 g, of which 37.6 g was of bones, 28.5 g of skin, and 5.77 g of dissectible fat. The average meat to bone ratio was 6.37.

### 3.2. Physico-Chemical Meat Quality

One of the objectives of the present study was to investigate whether 4% SWM-DEF dietary administration into different stages of broiler’s productive life could influence the meat physico-chemical properties (Table 6 and Table 7).

Results of the present experiment indicated that the dietary treatment was not a relevant factor since chicken breast pHu, as well as drip, thawing, cooking, and overall losses were similar in the three experimental groups (*p* > 0.05), and the same was observed for chicken leg traits, which included the femur resistance to fracture too. The sole exceptions were breast thawing loss and leg drip loss. Breast of C and SWM1 treatments highlighted a higher drip loss compared to SWM2 (*p* < 0.05). Legs of the SWM1 and SWM2 groups displayed a higher moisture loss than C one (2.44 and 2.16 vs. 1.59% for SWM2, SWM1 and C, respectively; *p* < 0.01).

Results about the proximate composition of broiler’s breast and leg meat (Table 7), highlighted that the SWM-DEF inclusion had an effect on the considered traits. Protein amount was affected both in the breast (*p* < 0.001) as well as in the leg (*p* < 0.01) meat cuts. Furthermore, ash content showed different values in the three groups for the breast meat cut (*p* < 0.01). In detail, SWM2 displayed a similar protein amount than C and both were higher than SWM1. This was observed both for breast (23.1 and 22.8 vs. 22.0 g/100 g for SWM2, C and SWM1 groups, respectively) and leg (20.0 and 20.0 vs. 19.5 g/100 g for SWM2, C and SWM1 groups, respectively) meat. Regarding ash content of breast meat, SWM2 showed a higher amount compared to SWM1 and C, the latter being similar (1.26 vs. 1.16 and 1.18 g/100 g for SWM2, SWM1 and C groups, respectively).

## 4. Discussion

The effect of silkworm meal on the productive performance of poultry was assessed in different studies [14,17,29,30]. However, findings are somewhat contrasting and this is attributable to the fact that silkworm species (*Bombyx mori* vs. *Samia ricini*), inclusion level and product type (i.e., full-fat, defatted), poultry species, and productive type (broiler chicken, broiler quail, layer chicken, etc.) differ in studies.

To cite some examples, Pietras et al. [17] found results similar to our study when the combination of silkworm pupae meal (17% dietary inclusion) and lupine seeds replaced soybean meal in the diet for growing chickens. The weight gains of the present trial (6 weeks) were comprised between 55 and 60 g/day/bird which is in line with the results of Khan et al. [16], where an average growth of 58.2 g/day/bird in a 6-week period was recorded. Similar outcomes were also highlighted in the study by Ijaiya and Eko [15]. Conversely, other studies reported that different inclusion levels of silkworm meal improved chickens’ live performance (7.8% and 2, 4, and 6%, for [16,31], respectively), while a 100% soybean meal replacement with silkworm meal worsened the live performance of chickens compared to the control diet [32].

The feed consumption directly affects the weight gain and thereby the live weight. It was interesting to notice that, no matter the SWM-DEF inclusion period, feed consumption was similar in all groups which was in line with the work by Miah et al. [14] on Rhode Island Red x Fayoumi crossbred chickens, where 50% soybean was replaced by full-fat silkworm meal, and by Ullah et al. [32]. This is of particular relevance since a feed-choice trial conducted by Dalle Zotte et al. [18] observed that meat-producing quails in their growth stage disliked the dietary inclusion of either full or defatted silkworm in the dose of 12.5% and, in fact, nutrient digestibility was also negatively affected. The explanation could be either dose-dependent or species-dependent, the latter being factors yet to be considered. An aspect that must be considered when including SWM in the diet of any animal species is the presence of the bio-compounds chitin and, especially, 1-Deoxynojirimycin (1-DNJ), which was shown to negatively affect nutrients digestibility [18,19], thus having an impact on live performance.

All the above-cited literature on feeding trials including SWM focused on one or the global chickens’ growth period, while none of them considered the possible role of the feeding stage on productive outcomes. The latter is of utmost importance, since different poultry growth stages could imply a different ability of the bird to digest, and thus to utilize SWM nutrients for growth, therefore being a key productive factor to consider. Looking at the live performance of chickens, results of the present trial seem to indicate that, at 4% inclusion level, SWM-DEF can be included in broiler diets either in the starter or grower-finisher periods without any negative effect on live performance, slaughter and carcass traits. This fact demonstrates that SWM-DEF is a promising protein source for young (1–10 days of life) and grower-finisher (11–42 days of life) chickens.

Results of the present study highlighted that carcass, breast, and leg yields were not affected by the dietary inclusion of 4% SWM-DEF which is in line with findings of other studies on silkworm meal as protein source for chickens [14,15,16,17,32,33,34]. The same was not always observed when different insect species (*Hermetia illucens, Tenebrio molitor, Musca domestica*) were tested [16], which might be attributable to different protein quality and/or efficiency of amino acid utilization. However, further research is needed to corroborate this hypothesis.

Physico-chemical characteristics of meat are relevant aspects for meat producers, to identify possible challenges and to adopt optimum processing strategies, and for consumers, since product quality has a direct impact on health [35]. A study testing the dietary inclusion of full-fat silkworm meal into Rhode Island Red × Fayoumi crossbred chickens’ diet [14] observed a pHu increase, which was hypothetically attributed to the hypoglycemic activity of bioactive compounds present in silkworm, which would result in a low lactic acid accumulation in muscles after slaughter and consequent higher pHu. However, in the present study breast and leg pHu of chickens was similar in all treatment groups, thus probably indicating that a 4% SWM-DEF inclusion level prevents this phenomenon to occur. Therefore, once again, the importance of identifying the optimum inclusion level for relatively unknown feedstuffs is emphasized.

To the best of our knowledge, this is the first study that reports the effects of silkworm meal dietary inclusion for chickens on the water holding capacity (WHC) of meat. Referring to studies dealing with other insect species, when soybean meal was substituted by partially defatted *Hermetia illucens* larval meal [36], 1.9% drip loss was registered in breast meat of both in experimental and control samples, thus not differing. Similarly, a 10% dietary inclusion with *Musca domestica* larvae meal showed no effect on chicken breast and thigh meat water holding capacity [37].

The WHC is a relevant property of meat which impacts product quality and it is intimately connected with pHu [38,39]. Results of the present trial showed that pHu was similar in both breast and leg meat in all treatments and, concurrently, the same was observed for total moisture loss (drip + thawing + cooking loss). Despite this, SWM-DEF was shown to exert some effect on the WHC of chicken breast (thawing loss) and of leg (drip loss) meat. The higher drip loss values of SWM1 and SWM2 leg meat compared to C could be linked to a different fatty acid profile of meat (not assessed in the present research): since silkworm pupa is known to be a rich source of omega-3 fatty acids [13], membrane damages due to oxidative phenomena could hypothetically have been accelerated in SWM1 and SWM2 groups, thus leading to a quicker moisture loss. This aspect however needs to be appropriately investigated, also considering the fact that the same was not observed in breast meat. Another possible explanation for the above-mentioned results could be linked to the observed differences in the proximate composition (especially protein content) of chicken breast and leg meat depending on the dietary group.

The differences highlighted in the proximate composition of chicken breast and leg meat could be due to freezing and frozen storage, together with a different fatty acids profile of meat resulting from the dietary treatment. Specifically, the highest ash content of the SWM2 breast meat could be explained by the lowest thawing loss value of the same group. Furthermore, freezing and thawing processes can lead to oxidative modifications of the amino acid side chains of proteins, which is initiated by the peroxidation of the polyunsaturated fatty acids, and leads to structural changes in the proteins which become less soluble [40]. Another hypothesis, the most likely, to explain the different proximate composition of SWM1 dietary treatment compared to the others, is that feeding SWM-DEF in the starter phase penalizes muscle tissue development. In fact, it is known that quality and quantity of dietary protein affect muscle mass, metabolism, and growth [41]. The observed result could thus be attributable to the known presence of anti-nutritional factors in the silkworm pupa (i.e., chitin and 1-DNJ) which, associated to the young age of chickens, does not allow the complete absorption and utilization of SWM-DEF protein.

## 5. Conclusions

Results of the present research highlighted that the dietary inclusion of 4% SWM-DEF into broiler chicken’s diet can be considered a realistic level: chickens evidenced satisfactory growth performance, slaughter and carcass traits, and physico-chemical meat quality. In addition, results suggested that the growth stage of chickens could be a relevant factor for the SWM-DEF dietary inclusion which needs to be further investigated; according to the meat proximate analysis results, the best option is to include SWM-DEF in the grower-finisher period, the choice being also dependent on cost-benefit considerations. Another aspect that could play a role in choosing the optimum inclusion period of SWM-DEF, and that will be discovered with further research, is the resulting fatty acids profile of the meat, since silkworm pupa is rich in omega-3 fatty acids which are important for human’s health. Furthermore, the fatty acid profile of meat, together with the assessment of the susceptibility to oxidation of its lipids, are of particular interest to shed some light on the results regarding WHC of meat.

## Figures and Tables

**Table 1 animals-13-00119-t001:** Ingredients of the experimental diets (g/kg feed).

Ingredients	Experimental Diets					
	C Starter	C Grower	C Finisher	SWM Starter	SWM Grower	SWM Finisher
Corn	310	342	396	329	360	411
Wheat	250	250	250	250	250	250
Soybean meal	370	330	280	316	277	230
Silkworm meal				40	40	40
Limestone	10	13	9	10	13	9
Sunflower oil	30	40	45	25	35	40
Starter premix ^1^	30			30		
Grower premix ^2^		25			25	
Finisher premix ^3^			20			20

^1^ Starter premix: Ca 4.13 g/kg, P 2.31 g/kg, Na 1.34 g/kg, Fe 48.6 mg/kg, Cu 16.5 mg/kg, Mn 120 mg/kg, Zn 110 mg/kg, I 1.2 mg/kg, Se 0.40 mg/kg, Vitamin-A 15180 I.U./kg, Vitamin-D3 5000 I.U./kg, Vitamin-E 85.8 I.U./kg, Lysine 2.18 g/kg, Methionine 2.79 g/kg (Producer: Agrofeed Ltd., Győr, Hungary); ^2^ Grower premix: Ca 2.55 g/kg, P 1.99 g/kg, Na 1.33 g/kg, Fe 48.8 mg/kg, Cu 16.3 mg/kg, Mn 120 mg/kg, Zn 110 mg/kg, I 1.2 mg/kg, Se 0.44 mg/kg, Vitamin-A 10,000 I.U./kg, Vitamin-D3 5000 I.U./kg, Vitamin-E 85.0 I.U./kg, Lysine 2.07 g/kg, Methionine 2.00 g/kg (Producer: Agrofeed Ltd., Győr, Hungary); ^3^ Finisher premix: Ca 2.50 g/kg, P 1.04 g/kg, Na 1.34 g/kg, Fe 48.6 mg/kg, Cu 15.0 mg/kg, Mn 120 mg/kg, Zn 110 mg/kg, I 1.2 mg/kg, Se 0.39 mg/kg, Vitamin-A 10,000 I.U./kg, Vitamin-D3 5000 I.U./kg, Vitamin-E 84.0 I.U./kg, Lysine 1.66 g/kg, Methionine 1.00 g/kg (Producer: Agrofeed Ltd., Győr, Hungary).

**Table 2 animals-13-00119-t002:** Chemical composition of the experimental diets (g/kg, as feed basis) and metabolizable energy content (MJ/kg feed).

Ingredients	SWM-DEF	Soybean Meal	Experimental Diets					
			C ^1^ Starter	C Grower	C Finisher	SWM ^2^ Starter	SWM Grower	SWM Finisher
Dry matter	947	920	918	917	916	919	921	917
Crude protein	597 ^3^	432	225	211	190	224	208	189
Ether extract	94.9	8.0	53	63	69	56	66	72
Crude fibre	-	63	32	30	31	30	30	28
Ash	66.5	68.5	61	54	46	62	59	49
Lysine	37.2	25.8	14.4	13.2	11.5	14.4	13.2	11.5
Methionine	19.8	5.8	6.21	5.20	3.92	6.66	5.66	4.40
Chitin	-	-	-	-	-	0.20	0.20	0.20
ME ^4^	21.9 ^5^	17.1 ^5^	12.5	12.9	13.3	12.5	12.9	13.3

^1^ C: commercial corn-soybean meal-based diet without insect meal inclusion; ^2^ SWM: 4% of defatted silkworm (*Bombyx mori* L.) meal replaced part of soybean meal; ^3^ N ∗ 5.60; ^4^ Metabolizable energy (MJ/kg feed); ^5^ Gross energy (MJ/kg feed).

**Table 3 animals-13-00119-t003:** Effect of the dietary inclusion either with 0% (Control) or 4% of defatted silkworm (*Bombyx mori* L.) meal during starter (SWM1) or grower and finisher (SWM2) phases on broilers’ growth performances.

Traits	C	SWM1	SWM2	RSD ^1^	*p*-Value
Animals, No.	30	30	30		
Live weight (g)					
1st day of age (start of trial)	39	39	39	2.2	0.384
10th day of age (end of starter period)	156	156	169	30.6	0.157
25th day of age (end of grower period)	911	973	1006	195	0.156
42nd day of age (end of finisher period)	2339	2498	2543	506	0.086
Weight gain (g/day/bird)					
Starter phase (1–10 days of age)	12	12	13	3.0	0.175
Grower (11–25 days of age)	50	54	56	11.5	0.155
Finisher (26–42 days of age)	84	90	91	15.4	0.184
Total (1–42 days of age)	55	59	60	9.4	0.109

^1^ Residual standard deviation.

**Table 4 animals-13-00119-t004:** Effect of the dietary inclusion either with 0% (Control) or 4% of defatted silkworm (*Bombyx mori* L.) meal during starter (SWM1) or grower and finisher (SWM2) phase on broilers feed intake and feed conversion ratio (FCR).

Traits	C	SWM1	SWM2	RSD ^1^	*p*-Value
Pens, No.	5	5	5		
Feed intake (g/day/bird)					
Starter phase (1–10 days of age)	24	25	22	2.8	0.322
Grower (11–25 days of age)	74	74	73	4.0	0.985
Finisher (26–42 days of age)	160	166	161	8.0	0.451
Total (1–42 days of age)	97	99	97	4.2	0.532
FCR ^2^ (kg/kg)					
Starter phase (1–10 days of age)	2.12	2.09	1.70	0.351	0.097
Grower (11–25 days of age)	1.47	1.36	1.32	0.138	0.169
Finisher (26–42 days of age)	1.90	1.87	1.89	0.153	0.900
Total (1–42 days of age)	1.77	1.70	1.68	0.106	0.397

^1^ Residual standard deviation; ^2^ FCR = feed conversion ratio (kg/kg).

**Table 5 animals-13-00119-t005:** Effect of the dietary inclusion either with 0% (Control) or 4% of defatted silkworm (*Bombyx mori* L.) meal during starter (SWM1) or grower and finisher (SWM2) phase on carcass traits and chicken leg dissection yields.

Traits	C	SWM1	SWM2	RSD ^1^	*p*-Value
Carcass No.	15	15	15		
Slaughter weight (SW), g	2519	2610	2719	282	0.1637
Carcass weight (CW), g	1722	1808	1873	214	0.1630
Carcass yield, %SW	68.3	69.2	68.8	1.59	0.2775
Leg weight, g	515	528	551	69.8	0.3716
Leg yield, %CW	29.8	29.2	29.4	1.49	0.5390
Breast weight, g	517	547	579	80.8	0.1244
Breast yield, %CW	30.1	30.2	30.9	2.33	0.6086
Leg No. ^2^	10	10	10		
Weight (LW) with skin, g	270	266	292	36.8	0.2622
Meat weight, g	235	228	252	33.7	0.2912
Bones weight, g	35.5	37.5	39.9	4.75	0.1405
Bones weight, %LW	13.2	14.1	13.8	1.42	0.3423
Meat to bone ratio	6.70	6.11	6.31	0.78	0.2512
Skin weight, g	28.2	28.7	28.5	7.35	0.9897
Skin weight, %LW	10.5	10.7	9.71	2.06	0.5148
Dissectible fat, g	7.36	4.73	5.22	2.83	0.1057
Dissectible fat, %LW	2.76	1.80	1.82	1.01	0.0664

^1^ Residual standard deviation; ^2^ One leg.

**Table 6 animals-13-00119-t006:** Effect of the dietary inclusion either with 0% (Control) or 4% of defatted silkworm (*Bombyx mori* L.) meal during starter (SWM1) or grower and finisher (SWM2) phase on pHu, drip loss, thawing loss and cooking loss of chicken breast and leg meat, and femur bone Warner–Bratzler Fracture Toughness (WBFT).

Traits	C	SWM1	SWM2	RSD ^1^	*p*-Value
Samples, No.	15	15	15		
Breast:					
pH_u_	6.10	6.01	6.02	0.23	0.4892
Drip loss, %	3.71	3.50	3.95	1.41	0.6819
Thawing loss, %	13.7 ^a^	13.7 ^a^	11.1 ^b^	2.83	0.0193
Cooking loss, %	31.2	33.8	32.5	4.54	0.3044
Total loss, %	48.7	51.0	47.6	5.00	0.1706
Leg:					
pH_u_ (*Biceps femoris*)	6.39	6.36	6.37	0.22	0.9446
Drip loss, %	1.59 ^Bb^	2.16 ^ABa^	2.44 ^A^	0.63	0.0022
Thawing loss, %	4.88	5.88	5.90	1.62	0.1583
Cooking loss, %	32.7	32.4	30.9	3.50	0.3394
Total loss, %	39.2	40.4	39.3	4.63	0.7135
Femur WBFT, N	357	357	391	57.5	0.3179

^1^ Residual standard deviation; ^A,B^ Means in the same row with different superscript letters differ for *p* < 0.01; ^a,b^ Means in the same row with different superscript letters differ for *p* < 0.05.

**Table 7 animals-13-00119-t007:** Effect of the dietary inclusion either with 0% (Control) or 4% of defatted silkworm (*Bombyx mori* L.) meal during starter (SWM1) or grower and finisher (SWM2) phase on the proximate composition (g/100 g) of chicken breast and leg meat.

Traits	C	SWM1	SWM2	RSD ^1^	*p*-Value
Breast meat					
Samples, No.	15	15	15		
Moisture	74.9	75.2	74.6	0.82	0.1638
Protein	22.8 ^ABa^	22.0 ^Bb^	23.1 ^A^	0.70	0.0003
Lipids	2.06	1.99	1.97	0.33	0.7576
Ash	1.18 ^B^	1.16 ^B^	1.26 ^A^	0.05	<0.001
Leg meat					
Samples, No.	15	15	15		
Moisture	76.3	76.8	76.4	0.76	0.1223
Protein	20.0 ^A^	19.5 ^B^	20.0 ^A^	0.36	0.0022
Lipids	3.52	3.39	3.20	0.66	0.4008
Ash	1.09	1.10	1.09	0.04	0.9896

^1^ Residual standard deviation; ^A,B^ Means in the same row with different superscript letters differ for *p* < 0.01; ^a,b^ Means in the same row with different superscript letters differ for *p* < 0.05.

## Data Availability

Data are available upon request to the corresponding author.

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
