# Peer review of "Dietary Inclusion of Defatted Silkworm (Bombyx mori L.) Pupa Meal for Broiler Chickens at Different Ages: Growth Performance, Carcass and Meat Quality Traits"

_animals, 2022, doi:10.3390/ani13010119_

Round 1

Reviewer 1 Report

This manuscript evaluates the inclusion of 4% defatted silkworm pupa meal  for broiler chickens for 2 different broiler feeding periods. Suggestions and comments are provided for consideration by line numbers below.

L. 30 to 32  Consider rephrasing. I do not think you measured pen feed intake on day 1 and I assume your measurements were made after each period, not before.

L. 34 Consider rewording this statement (Results confirmed that silkworm is an interesting protein source . . .). You showed no significant differences. That really means that you were not able to demonstrate differences. It does not demonstrate that performance was the same.

L. 36 to 37  Is reduced crude protein of the mean a benefit?

L. 37 to 40  I am not sure how you can conclude this. You did not do a cost-benefit analysis. In addition, was the nutritional quality of the meat improved? I would argue that reducing protein content is negative and you did not see any other differences. Plus SWM-DEF inclusion increased drip loss in leg, which is a negative.  I suggest to rewrite this section, also considering the comment for L. 34.

L. 81 and 84  You state that it is important to determine the proper inclusion level, but then pick only 1 level to study. I understand limitations on the number of treatments that can be used. It may be beneficial to briefly indicate why you picked 4%. Was it based on optimal levels in other studies? Please clarify.

L. 97 to 99  I am not sure about the rules in other countries, but I have to have approved protocols by the animal care committee if it involves research, even if it is just feeding and weighing. It may be better to state that approval was not deemed necessary by the local ethics committee (as suggested in the letter you provided)

L. 112  You have day 1 to 10 here, but 0 to 11 in the abstract and 1st day and 10th day in the Table. Please correct throughout. Same for the next period and the overall period. Please check for accuracy and consistency.

L. 115 and 116  I think it is important to explain how you formulated the diets. First, I assume you formulated based on CP content of SWM-DEF and SBM. What was the CP content of SWM-DEF? What was it for SBM? Did you analyze the ingredients? My next question is about amino acid composition. Most commercial nutritionists do not formulate based on CP, but the formulate based on amino acids (true digestible). I do not know the amino acid composition of SWM-DEF, so I can not gauge if there was an amino acid balance issue. This is an important consideration when interpreting the results. You also indicate the diets were isocaloric. What energy value did you use for SWM-DEF? If you have the analyzed composition of the SWM-DEF, you should include the data. I think it would add tremendously to the manuscript. Please add more details regarding “assumptions” and perspectives in diet formulation.

L. 116 and

L. 131 Delete the extra comma

L. 130 to 133  Use parenthesis to show abbreviations, not dashes.

L. 149, 160, and 169  There is really not a need to include formulas here. It is logical to state (as you have) that e.g. drip loss is expressed relative to the original sample weight. I suggest deleting the formulas in this section.

L. 183 to 189  I think that this section would be better placed after you discuss the diets? More importantly, I do not understand why you analyzed the diets and then reported calculated values. The authors really should report the analyzed values, which also serves as a double-check that assumptions for formulating the diets were accurate. Having calculated values is o.k., but you should include analyzed values as well.

L. 191 to 196  Please indicate your experimental unit. I assume it is pen, but from the Tables, it looks like you used individual bird for growth and pen for feed intake and efficiency. Technically, diets were applied to pens and pen really should be experimental using for growth too.

L. 191 to 196  You state that post-hoc pairwise comparison was performed using the Bonferroni correction, but you do not indicate that in the data; I presume you do that only if the treatment effect is significant. Indeed, it looks like you show P values for overall treatment effects. I am not sure why you report RSD rather than SEM; I am more used to seeing SEM in these types of data and I find it easier to interpret.

Table 3  Just a comment.  It would be interesting to know what the P value is for the difference between C and SWM2 for d 42 weight (for example). It is encouraging that performance results are at least numerically better with SWM and that supports your argument that no difference was detected.

L. 298 You may want to say what the different inclusion levels were to provide perspective.

L. 311 Did you measure the 1-Deoxynojirimycin concentration in the product you used? If so, it would be good to include.

L. 320  To state “at any stage” is not accurate. You did 2 periods.

L. 377  It is interesting to note that the feeding period during the started was 10 days and the other feeding period was 32 days. So, thinking about total consumption of SWM-DEF in a young broiler (low feed intake and fed for 10 days) and a larger bird (eating more plus fed for 32 days) would imply that the older bird should be impacted more (ie. more n-3 FA accumulation), even more so because older birds would deposit more fat as well. Just a thought.

Throughout the manuscript, there are some grammar and wording issues that should be fixed. I did not specifically identify those in this review. Please carefully edit your manuscript.

Author Response

Manuscript ID: animals-2088572

Title: Dietary inclusion of defatted silkworm (Bombyx mori L.) pupa meal for broiler chickens at different ages: growth performance, carcass and meat quality traits

First of all, Authors would like to express their gratitude to the contribution provided by the reviewers to improve the quality of the present manuscript. All suggestions and comments have been carefully considered and a point by point replies have been provided. All the responses and text corrections are in red font.

Reviewer #1

This manuscript evaluates the inclusion of 4% defatted silkworm pupa meal for broiler chickens for 2 different broiler feeding periods. Suggestions and comments are provided for consideration by line numbers below.

L. 30 to 32: Consider rephrasing. I do not think you measured pen feed intake on day 1 and I assume your measurements were made after each period, not before.

The Reviewer is right and Authors have now rephrased the sentence (now L31-32).

L. 34: Consider rewording this statement (Results confirmed that silkworm is an interesting protein source . . .). You showed no significant differences. That really means that you were not able to demonstrate differences. It does not demonstrate that performance was the same.

R: Authors have rephrased the sentence (L34-36).

L. 36 to 37: Is reduced crude protein of the mean a benefit?

R: Definitively not, authors have now completed the sentence improving clarity (L38).

L. 37 to 40: I am not sure how you can conclude this. You did not do a cost-benefit analysis. In addition, was the nutritional quality of the meat improved? I would argue that reducing protein content is negative and you did not see any other differences. Plus SWM-DEF inclusion increased drip loss in leg, which is a negative. I suggest to rewrite this section, also considering the comment for L. 34.

R. Maybe author’s message has not been correctly delivered in the text and authors are sorry about this. What authors wanted to emphasize is that since SWM1 and SWM2 diets ensured satisfactory performance (not different from those obtained with a commercial diet - control) and overall meat quality was also satisfactory (the only exceptions were: drip loss which was 0.57-0.85 % higher in SWM meat compared to Control and protein content which was 0.8-0.9 % lower in SWM1 compared to SWM2 and Control), the choice of the best inclusion period, with the results available up to now, is a mostly a matter of cost-benefit consideration (i.e. it is the company’s/producer’s that should choose the best option based on cost-benefit considerations). Authors have modified the conclusions to be more clear.

L. 81 and 84: You state that it is important to determine the proper inclusion level, but then pick only 1 level to study. I understand limitations on the number of treatments that can be used. It may be beneficial to briefly indicate why you picked 4%. Was it based on optimal levels in other studies? Please clarify.

R. As the Reviewer’s correctly wrote, the choice to test one inclusion level was due to structural limitations of the poultry farm. However, of course there is a scientific reason behind the choice of a 4% inclusion level and authors have now highlighted this aspect at L 74-76.

L. 97 to 99: I am not sure about the rules in other countries, but I have to have approved protocols by the animal care committee if it involves research, even if it is just feeding and weighing. It may be better to state that approval was not deemed necessary by the local ethics committee (as suggested in the letter you provided)

R. Authors followed the Reviewer’s suggestion and specified this aspect at L 101-102.

L. 112: You have day 1 to 10 here, but 0 to 11 in the abstract and 1stday and 10thday in the Table. Please correct throughout. Same for the next period and the overall period. Please check for accuracy and consistency.

R. Throughout the manuscript, corrections have been made wherever necessary.

L. 115 and 116: I think it is important to explain how you formulated the diets. First, I assume you formulated based on CP content of SWM-DEF and SBM. What was the CP content of SWM-DEF? What was it for SBM? Did you analyze the ingredients? My next question is about amino acid composition. Most commercial nutritionists do not formulate based on CP, but the formulate based on amino acids (true digestible). I do not know the amino acid composition of SWM-DEF, so I can not gauge if there was an amino acid balance issue. This is an important consideration when interpreting the results. You also indicate the diets were isocaloric. What energy value did you use for SWM-DEF? If you have the analyzed composition of the SWM-DEF, you should include the data. I think it would add tremendously to the manuscript. Please add more details regarding “assumptions” and perspectives in diet formulation.

R. Authors have improved the description of experimental diets formulation at L119-122. The proximate composition of SWM-DEF and soybean meal have now been included in Table 2. Authors have also included the Lysine and Methionine content of the SWM-DEF, soybean meal and experimental diets; however, we cannot provide the complete amino acid profile as values are included in a second manuscript which is dealing also with the amino acid profile of meat. Regarding feed formulation, it was carried out by Agrofeed Ltd (Hungary) and it considered metabolizable energy and digestible amino acids.

L. 116: and

R. Authors have revised and inserted comma.

L. 131: Delete the extra comma

R. Deleted.

L. 130 to 133: Use parenthesis to show abbreviations, not dashes.

R. Corrected.

L. 149, 160, and 169: There is really not a need to include formulas here. It is logical to state (as you have) that e.g. drip loss is expressed relative to the original sample weight. I suggest deleting the formulas in this section.

R. Authors followed the recommendations and eliminated the formulas as reference are also provided.

L. 183 to 189: I think that this section would be better placed after you discuss the diets? More importantly, I do not understand why you analyzed the diets and then reported calculated values. The authors really should report the analyzed values, which also serves as a double-check that assumptions for formulating the diets were accurate. Having calculated values is o.k., but you should include analyzed values as well.

R. Authors have anticipated the position of this paragraph to 2.2, and in Table 2 have now reported analysed values only.

L. 191 to 196: Please indicate your experimental unit. I assume it is pen, but from the Tables, it looks like you used individual bird for growth and pen for feed intake and efficiency. Technically, diets were applied to pens and pen really should be experimental using for growth too.

R. Revised and modified according to the reviewer’s comment (now L195-198).

L. 191 to 196: You state that post-hoc pairwise comparison was performed using the Bonferroni correction, but you do not indicate that in the data; I presume you do that only if the treatment effect is significant. Indeed, it looks like you show P values for overall treatment effects. I am not sure why you report RSD rather than SEM; I am more used to seeing SEM in these types of data and I find it easier to interpret.

R. RSD is highly related to SEM (SEM=RSD/n, where n is the number of observations with which the corresponding LS-means is calculated). In this case, the 3 groups are numerically balanced so that RSD is just as informative as SEM.

Table 3: Just a comment. It would be interesting to know what the P value is for the difference between C and SWM2 for d 42 weight (for example). It is encouraging that performance results are at least numerically better with SWM and that supports your argument that no difference was detected.

R. The specific P value for the difference between C and SWM2 for d42 live weight is 0.083.

L. 298: You may want to say what the different inclusion levels were to provide perspective.

R. Authors have now provided the P-values for the cited references at L302.

L. 311 Did you measure the 1-Deoxynojirimycin concentration in the product you used? If so, it would be good to include.

R. Unfortunately authors did not have the possibility to analyse DNJ concentration for the present study.

L. 320: To state “at any stage” is not accurate. You did 2 periods.

R. Authors have corrected the text accordingly (L324-325).

L. 377: It is interesting to note that the feeding period during the started was 10 days and the other feeding period was 32 days. So, thinking about total consumption of SWM-DEF in a young broiler (low feed intake and fed for 10 days) and a larger bird (eating more plus fed for 32 days) would imply that the older bird should be impacted more (ie. more n-3 FA accumulation), even more so because older birds would deposit more fat as well. Just a thought.

R. Authors fully agree with this thought; in relation to this aspect, authors are now busy analysing the data about the FA profile and content of chicken meat obtained from SWM1 and SWM2 treatments.

Throughout the manuscript, there are some grammar and wording issues that should be fixed. I did not specifically identify those in this review. Please carefully edit your manuscript.

R. Authors have thoroughly revised the manuscript for grammatical and wording mistakes.

Reviewer 2 Report

This experiment used defatted silkworm as a partial replacement to SBM in broiler diet. I agree that it is a hot topic in poultry and this research provides a some novel contribution to the use of insect as a feed for poultry. Overall, the experimental methods and design were conduct correctly and the manuscript is well written with sufficient presentations of results and discussion. I have several suggestions and comments after thoroughly reviewing the manuscript, as follows:

-        Authors said that it is fundamental to establish satisfactory inclusion level; however they only use one inclusion level. Thus, justification regarding the inclusion level they used should be provided.

Materials and methods

-        How the rations were made/mixed? Please indicate!

-        Statistical analysis: please declare the random effect of the stats!

-        (L195): I don’t think benferroni is the best fit for post-hoc comparison, it does not compare among groups, instead only two groups were compared. Please revisit and re-do your stats using Tukey to compare the least square means among three groups of your study! There is a potential to be significantly different between control and SWM2 groups in some parameters…

-        I would be great if you have the chemical composition of the SWM so people know the nutritional profile of the SWM!

-        Please add the Lys content of the rations!

Results:

-        Given the LSM of the live weight and FCR (starter and total phase) with their respective RSD, you should find it significantly different using Tukey test, please check! If so, please adjust your results! Difference in 0.9 (1.77 vs 1.68) point of FCR means a lot for industrial feed cost perspective!

Conclusion:

-        L387: I would suggest the authors to remove this sentence regarding the FA profile. In fact, the SWM is defatted and I would not expect it would contribute to the FA profile improvement. But, directing out that for future study using fat/oil from the worm may be a correct recommendation. 

Author Response

Manuscript ID: animals-2088572

 Title: Dietary inclusion of defatted silkworm (Bombyx mori L.) pupa meal for broiler chickens at different ages: growth performance, carcass and meat quality traits

First of all, Authors would like to express their gratitude to the contribution provided by the reviewers to improve the quality of the present manuscript. All suggestions and comments have been carefully considered and a point by point replies have been provided. All the responses and text corrections are in red font.

Reviewer #2

This experiment used defatted silkworm as a partial replacement to SBM in broiler diet. I agree that it is a hot topic in poultry and this research provides a some novel contribution to the use of insect as a feed for poultry. Overall, the experimental methods and design were conduct correctly and the manuscript is well written with sufficient presentations of results and discussion. I have several suggestions and comments after thoroughly reviewing the manuscript, as follows:

Authors said that it is fundamental to establish satisfactory inclusion level; however they only use one inclusion level. Thus, justification regarding the inclusion level they used should be provided.

R. The choice to test one inclusion level was due to structural limitations of the poultry farm. However, of course there is a scientific reason behind the choice of a 4% inclusion level and authors have now highlighted this aspect at L 74-76.

Materials and methods:

How the rations were made/mixed? Please indicate!

R. Diets were formulated by Agrofeed Ltd and manufactured at the Animal Research Plant of the University. The information is available at L122-123.

Statistical analysis: please declare the random effect of the stats!

R. Revised accordingly (L195-196).

L195: I don’t think benferroni is the best fit for post-hoc comparison, it does not compare among groups, instead only two groups were compared. Please revisit and re-do your stats using Tukey to compare the least square means among three groups of your study! There is a potential to be significantly different between control and SWM2 groups in some parameters…

R. Bonferroni is the most widely used method for performing all pairwise comparisons between averages of three or more levels of a factor. It is also much more conservative than Tukey. Finally, in our work we are not only interested in making a comparison with the control but also between the two levels SWM1 and SWM2, so Bonferroni is the preferred test. Authors have however also performed again the stats applying the Tukey test and, despite P-values obviously slightly changed, no new significance was highlighted.

I would be great if you have the chemical composition of the SWM so people know the nutritional profile of the SWM!

R. The proximate composition of the SWM-DEF has been included in Table 2.

Please add the Lys content of the rations!

R. The Lysine and Methionine contents of the rations have been included in Table 2.

Results:

Given the LSM of the live weight and FCR (starter and total phase) with their respective RSD, you should find it significantly different using Tukey test, please check! If so, please adjust your results! Difference in 0.9 (1.77 vs 1.68) point of FCR means a lot for industrial feed cost perspective!

R. Authors fully agree with the reviewer comment about the relevant numerical difference in FCR. However, that was only a numerical difference as we have also applied the Tukey test and the P-value was not significant (P=0.399).

Conclusion:

L387: I would suggest the authors to remove this sentence regarding the FA profile. In fact, the SWM is defatted and I would not expect it would contribute to the FA profile improvement. But, directing out that for future study using fat/oil from the worm may be a correct recommendation. 

R. Authors thank the reviewer for the suggestion. However, authors would like to highlight that the defatted SWM used in the present study should still be considered a source of lipids as well (9.49% lipids as it is now reported in Table 2), and thus omega-3 PUFAs. For this reason, the authors did not modify the conclusion section.

Reviewer 3 Report

Dear Editor,
Here is my review on the manuscript with the number Animals-2088572.
Thanks for the cooperation.
Regards

Author Response

Manuscript ID: animals-2088572

Title: Dietary inclusion of defatted silkworm (Bombyx mori L.) pupa meal for broiler chickens at different ages: growth performance, carcass and meat quality traits

First of all, Authors would like to express their gratitude to the contribution provided by the reviewers to improve the quality of the present manuscript. All suggestions and comments have been carefully considered and a point by point replies have been provided. All the responses and text corrections are in red font.

Reviewer #3

L26: Animal number is low

R. Authors are conscious about this limit but 90 was the maximum number of replicated pens and animals that could be used in this experiment.

L27: What are initial body weights?

R. Initial body refers to the live weight of 1 day old chicks, therefore authors thought that it was not necessary to state it in this section as no difference were expected (weight of day-old chicks is strictly correlated with egg weight. Therefore, since the genetic source and incubator were the same, a relevant difference in day-old chicks live weight was fairly possible). Anyway, the live weight of day-old chicks can be found in Table 3.

L42: Keywords must be sorted alphabetically.

R. Authors did not sort keywords alphabetically as in the journal guidelines (instructions for authors) it is only mentioned: “Keywords: Three to ten pertinent keywords need to be added after the abstract. We recommend that the keywords are specific to the article, yet reasonably common within the subject discipline, https://www.mdpi.com/journal/animals/instructions”. However, authors have now sorted them alphabetically as requested.

L44: Add more information

R. The introduction provides the context (also legislative) on the use of insects as animal feed, then specifically presenting the Bombyx mori and research available up to now on poultry with positive-negative aspects and parts that still need to be elucidated. Finally, based on the premises the objective of the research has been indicated. Since the comment of the reviewer is generic, author don’t know to which section of the introduction is the reviewer referring to.

L52: Old references should be taken out.

R. A reference should not be deleted (not considered) only because it is old. When citing literature, it is the quality of the reference that matters. A cited paper can be of the 60’s but if it was a reference paper that it is still valid today, why should we forget it and cite someone else that used the same method in a paper of the 2010’s? Then, sometimes only few studies are available on a topic and some of them were published some decades ago. However, they should be considered anyway since they represent the sole available scientific data.

L71: Old references should be taken out.

R. The same comment as above applies

L100: What are initial body weights?

R. Initial body refers to the live weight of 1 day old chicks, therefore authors thought that it was not necessary to state it in this section as no difference were expected (weight of day-old chicks is strictly correlated with egg weight. Therefore, since the genetic source and incubator were the same, a relevant difference in day-old chicks live weight was fairly possible). Anyway, the live weight of day-old chicks can be found in Table 3.

L115-116: Which nutritional programs were used? NRC or another?????

R. The 2019 AVIAGEN guidelines for ROSS 308 have been followed, including the nutritional requirements and this information has now been included at L111-112.

L125-128: Give relevant reference

R. Since it was a commercial abattoir, the standard method was applied: Council Regulation, EC, No. 1099 (2009). This information has now been included at L144-145.

L153-157: Give relevant reference

R. A reference has been included for all moisture loss determinations.

L286: The discussion should be supported with more relevant information from the literature. ADD more references

R. The cited literature is all the literature of satisfactory scientific merit that is available on the topic. Does the reviewer have a specific suggestion on this?

L333: Old references should be taken out.

R. Please see the previous comments on this aspect.

L349: Don’t start with abbreviations, and You should write in full if use for the first time.

R. Revised according to the reviewer’s comment.

L373: Old references should be taken out.

R. Please see the previous comments on this aspect. Specifically, this is a key review on how the quality and quantity of dietary protein affects muscle mass, metabolism and growth in animal.
